



# Towards Bedmap Himalayas: a new airborne glacier thickness survey in Khumbu Himal, Nepal.

Hamish D. Pritchard[1], Edward C. King[1], David J. Goodger[1], Douglas Boyle[1,2], Daniel N.
Goldberg[3], Beatriz Recinos[3], Andrew Orr[1], Dhananjay Regmi[4]

[1]British Antarctic Survey, Cambridge, CB3 0ET, UK
[2]University of Cambridge, UK
[3]School of GeoSciences, University of Edinburgh, UK
[4]Himalayan Research Centre, Kathmandu, Nepal

*Correspondence to*: Hamish D. Pritchard (hprit@bas.ac.uk)

**Abstract**

Mountain glaciers provide an important service in sustaining river flows for large populations downstream of High Mountain Asia (HMA) but these glaciers are retreating and the future of this water resource is highly uncertain. Glacier thickness measurements are vital for accurate mapping of the remaining ice reserve and for predicting where and how fast it will decline under climate change, but such measurements are severely lacking in this region due to the difficulties of surveying in remote, high-altitude settings. We report on a uniquely extensive new thickness dataset for eleven glaciers in the Khumbu Himal around Mount Everest that we collected in late 2019 using a novel, low-frequency helicopter-borne radar. To aid in interpreting the survey radargrams we developed a terrain clutter model, and we succeeded in mapping ice thickness with a precision of around ±7 % and horizontal spacing of around 40 m, for thicknesses of up to 445 m and spanning a total of 119 line-km, approximately doubling the length of previous thickness surveys in HMA. To demonstrate the utility of our new measurements, we compare them to existing modelled thickness products and find that the models struggle to reproduce the distribution of ice in these complex, steep, rapidly slowing, thinning and stagnating glaciers, with widespread systematic thin and thick biases equivalent to around half of the measured ice thickness or more. This new dataset (https://doi.org/10.5285/e39647f5-fb72-4d16-acbd-9784ed2167b8) permits for the first time a detailed analysis of model performance on Himalayan glaciers, a key step in improving model skill and hence the accuracy of modelled thickness distributions and future ice loss on the mountain-range scale.

**Short summary**

We present a new and uniquely extensive dataset of glacier thickness from the Khumbu Himal around Mount Everest that stretches for 119 km, doubling the extent of thickness measurements in High Mountain Asia. Such measurements are key inputs for models that estimate how much ice is stored on the whole mountain range scale and for models that predict how this ice reserve will change in future, and what impact this will have on water supply for the large populations living downstream.

## 1 Introduction

High Mountain Asia (HMA) contains ~95,000 glaciers which provide an important service in sustaining river flow in the region's relatively warm and dry spring and autumn months, and particularly during droughts



(Pritchard 2019). As many of these glaciers are retreating due to climate change (Maurer, Schaefer et al. 2019),

major rivers including the Ganges, Indus and Brahmaputra are in the process of losing the protective hydrological buffer that their glacio-pluvial regimes provide. Large-scale mass balance modelling studies project that roughly half of the glacier ice in HMA (approximately 5,000 km³, enough to raise sea levels by over a centimetre (Huss and Farinotti 2012)) will be lost by 2100 but there is considerable uncertainty in these projections (Marzeion, Hock et al. 2020, Rounce, Hock et al. 2020). This is in part due to uncertainty in future climate, but uncertainty

in the glacier models themselves can match or exceed this climate uncertainty (Marzeion, Hock et al. 2020) for reasons that are intimately connected to a lack of ice thickness measurements (Li, Maussion et al. 2023).

Field surveys of ice thickness are sparse and insufficient on their own for mapping ice distribution beyond the local scale of the survey (e.g., Pritchard 2021), but in several ways they are of key importance to modelled

estimates of regional and global ice distribution (Millan, Mouginot et al. 2022) and projections of how this will evolve. In some simpler glacier models, thickness measurements are, for example, used to estimate the spatially distributed thickness distribution through empirical scaling relationships between glacier area, length and volume (Marzeion, Hock et al. 2020, Rounce, Hock et al. 2020). In more sophisticated dynamic ice flow models, they are used to constrain model properties such as ice stiffness and bed friction (e.g., Millan, Mouginot et al. 2022), and

such dynamic models have been employed to map regional and global ice thickness distribution through inversion of satellite-derived surface flow (Farinotti, Huss et al. 2019, Millan, Mouginot et al. 2022). These distributions have, however, been prone to large biases (Farinotti, Huss et al. 2019, Pritchard, King et al. 2020, Farinotti, Brinkerhoff et al. 2021, Millan, Mouginot et al. 2022).

Adding to this challenge to mapping the ice thickness distribution, the accuracy of forward modelling projections of glacier retreat under future climate change scenarios is also dependent on thickness-constrained model properties such as ice stiffness and bed friction, and on initial glacier extents, and thickness measurements are critical for refining these extents. Recent studies in HMA have shown that variations in extent between different inventories can cause significant differences in both present-day ice thickness estimates and future mass loss

projections, which can be even more sensitive to these inventory choices than to differences in climate forcing scenarios (Li, Maussion et al. 2023). More broadly, model initialisation errors relating to ice thickness cause spurious thinning and thickening patterns, biasing projections of mass change for decades or longer, and in HMA such initialisation uncertainty accounts for 50 % of overall glacier projection uncertainty in the first few decades of simulation (2020-2040) (Marzeion, Hock et al. 2020).


The assimilation of even limited measurements through calibrated inverse modelling techniques can markedly improve the accuracy of modelled projections of glacier retreat and loss, hence spatially distributed glacier thickness surveys are vital (Frey, Machguth et al. 2014, Farinotti, Huss et al. 2019, Maussion, Butenko et al. 2019, Marzeion, Hock et al. 2020, Jouvet and Cordonnier 2023), (Farinotti, Huss et al. 2019, Millan, Mouginot et al.

2022, Li, Maussion et al. 2023), particularly where the measurements sample the thickest parts of the target glaciers and not only the more accessible but typically thinner lower glacier elevations (Farinotti, Brinkerhoff et al. 2021). The profound lack of such measurements in HMA (Pritchard, King et al. 2020, Pritchard 2021) therefore





remains a key barrier to creating an accurate inventory of the region's current ice reserve, and to forecasting how this will change.


Ice thickness data are scarce primarily because they are difficult to obtain, even for individual HMA glaciers. Radar or seismic field surveys on the ground are hindered by the practical challenges of working on glaciers that are remote and lie at high altitude, are often very rough, debris-covered and crevassed, and are frequently struck by avalanches and rock falls from surrounding mountain slopes (e.g., Pritchard, King et al. 2020). Airborne radar

surveys by fixed-wing aircraft (commonly employed to measure the thickness of the polar ice sheets (e.g., Plewes and Hubbard 2001)) are hindered by a lack of manoeuvrability at high altitude, within the region's narrow and deeply incised glacial valleys. Furthermore, radar signal-penetration is hampered by the high water and debris content of the lower tongues of HMA glaciers (e.g., Macheret, Moskalevsky et al. 1993, Gades, Conway et al. 2000).


To overcome the practical challenges of surveying HMA glaciers, Pritchard, King et al. (2020) adapted a low-frequency ice-penetrating radar, developed originally for Antarctic over-snow surveys, for deployment by helicopter in areas characterised by thick, dirty, temperate ice. The light, portable, modular, low-frequency design of the Bedmap Himalayas radar platform, combined with the manoeuvrability of helicopters, make this approach

particularly well suited to surveying the remote and otherwise largely inaccessible high-mountain glaciers of this region. Here, we report on the first Himalayan survey with this platform, and the new glacier thickness dataset resulting from it.

## 2 Methods

### 2.1 Airborne radar survey

#### 2.1.1 Radar system

We used a wide band mono-pulse dipole radar with a centre frequency of 7 MHz (Pritchard, King et al. 2020) for this survey. The transmitter, receiving system, antennas and GPS were all mounted on a 24 m long semi-flexible, modular box-section structure built of polyester fibreglass tubing and non-metallic connectors, designed to be flown as a helicopter sling load (Figure 1). We employed transmit and receive antennas each 20 m in overall

length, comprising a pair of 10 m half-dipoles with resistor loading to suppress resonance. The antennas overlapped for 80 % of their length and were separated horizontally by 0.7m. This configuration kept the peak voltage entering the digitiser front-end below the overload limit but precluded the use of any amplification stage.

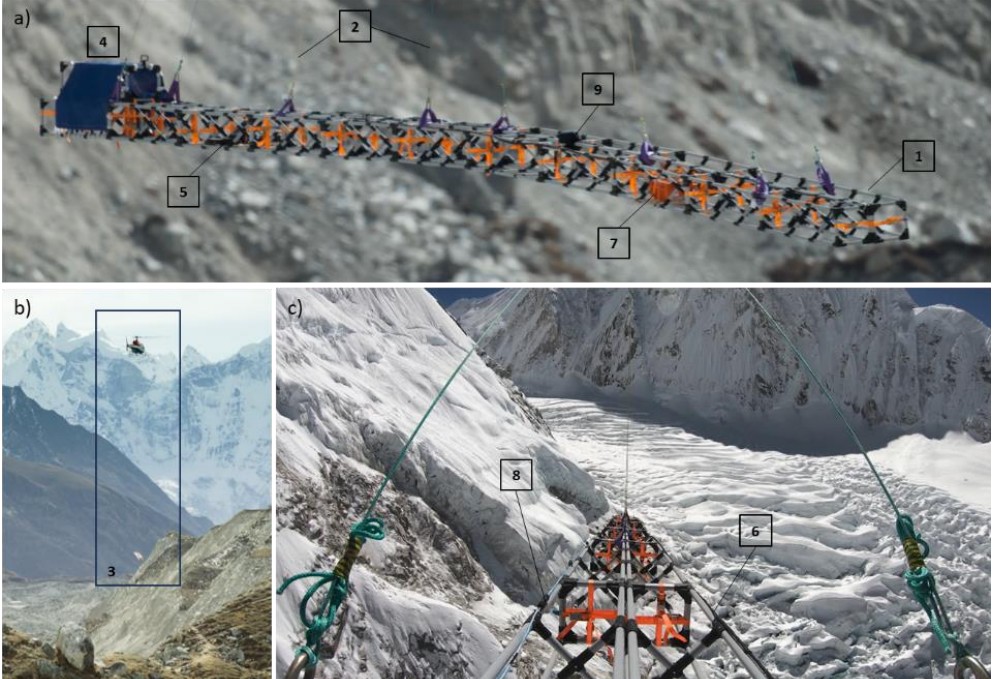

**Figure 1** | The radar frame consisting of lightweight fibreglass modules strapped together (1), with rope lines (2) to suspend the system under the helicopter (3), a tail (4) to prevent 'weathercocking', a pulse transmitter (5) and corresponding, resistively loaded transmit dipole antenna (6), receiver (7) and receive antenna (8) and GPS (9) (Pritchard, King et al. 2020). Panel (c) shows the system approaching the icefall on Khumbu Glacier (Figure 2), with multiple potential sources of radar clutter notably from the surrounding mountains and crevasse walls. Photo credits: Hamish Pritchard.

### 2.1.2 Airborne survey

From 2019-10-27 through 2019-11-06, we deployed our radar platform to survey the glaciers of Nepal's Khumbu Himal (Everest area) in the upper Dudh Koshi river basin, an area notable for holding some of the region's largest glaciers and highest accumulation areas (Figure 2). Porters transported on foot the fibreglass modules, rigging and radar components from the airport at Lukla to a small unpaved airstrip immediately above and north of Namche Bazaar, where we assembled the survey platform. From this staging site, we flew a series of test and survey helicopter flights over the glaciers shown in Figure 2 (see also Table A1). We used locally chartered AS350 helicopters and crew capable of high-altitude operations who operated out of Lukla Airport, and our survey flights covered >200 line-km spanning altitudes of 3700 m to 6700 m.

We flew our radar platform as an underslung load that transmitted continuously and required no electronic connection to the helicopter. The only mechanical connection was that between the helicopter sling hook and the platform rigging, a simple configuration that allowed the platform to be lifted and returned to the ground at our

staging site with or without the helicopter landing and which, for safety, allows the pilot to drop the platform in flight if needed (this was unnecessary during our survey).


Based on the findings of Pritchard, King et al. (2020), we designed our survey patterns to include multiple glacier cross profiles because these are less prone to ambiguity between radar returns from the glacier bed and valley side walls. Our flightlines typically followed a continuous zig-zag path with crossings spaced at ~800 m over each glacier trunk for the up-glacier survey limb, with these crossings subsequently linked by a central glacier long-

profile on descent. To minimise radar spreading losses, surveys were flown with the radar platform as close as safely possible to the glacier surface, typically a few metres to tens of metres given the considerable roughness of the glacier surfaces in this area. Reaching the highest section of the survey (Everest's Western Cwm), however, required a spiralling rather than direct ascent over the Khumbu Icefall and so achieved multiple glacier crossings at a wider range of ground clearances (Figure 2). To ensure dense radar sampling, we flew these glacier profiles

as slowly as was practicable (typically ~10 m s$^{-1}$ (36 km h$^{-1}$)). Transit flights to and from the glaciers were at higher speeds of up to ~40 m s$^{-1}$ (140 km h$^{-1}$). Our platform remained stable in flight and mechanically sound throughout the mission.

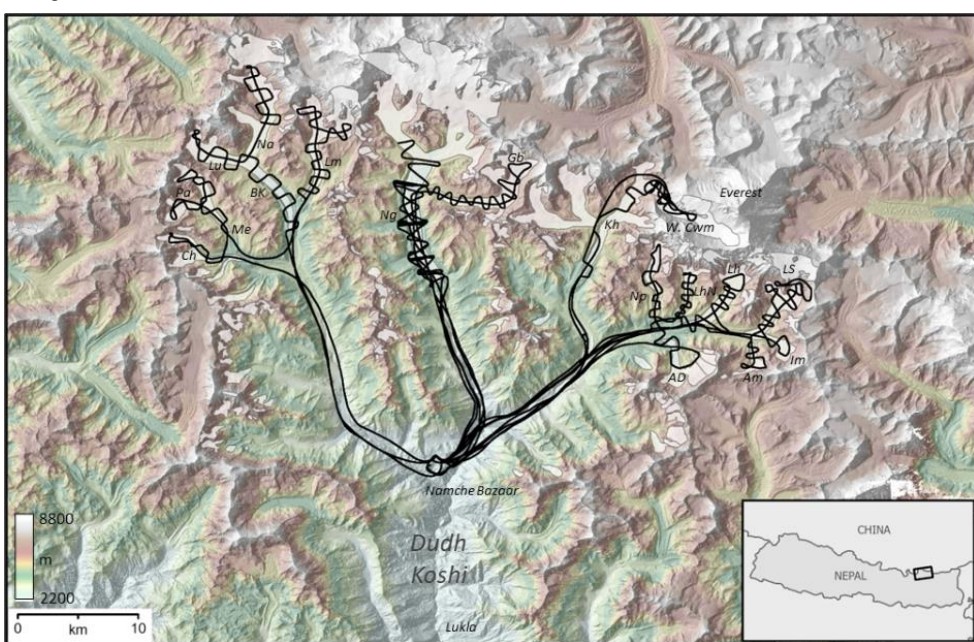

**Figure 2 |** Map of the airborne survey (black lines) over the glaciers of the Khumbu Himal, Nepal (white polygons

(RGI Consortium 2017)). Glacier names, after King, Quincey et al. (2017) and Realitymaps.de, are *Ch* = Chhule, *Me* = Melung, *Pa* = Pangbug, *BK* = Bhote Kosi, *Lu* = Lunag, *Na* = Nangpa, *Lm* = Lumsamba (Sumna), *Ng* = Ngozumpa, *Gb* = Gyubanare, *Kh* = Khumbu, *Np* = Nuptse, *Lh* = *Lhotse, LhN* = Lhotse Nup, *LS* = Lhotse Shar, *Im* = Imja, *AD* = Ama Dablam, *Am* = Ambulapcha, *W. Cwm* = Western Cwm of Everest. Background: coloured, shaded-relief topography (Jarvis, Reuter et al. 2008, Shean 2017).





### 2.2 Data processing

#### 2.2.1 Radar data processing

In processing our radar data, we aimed to produce a set of geometrically correct radargram images from our raw data that we could interpret manually to digitise (pick) profiles of the surfaces and beds of the glaciers, allowing us to quantify the ice thickness. We performed radar processing steps *b-i* using ReflexW version 10.1 software. The key steps were:

a. *Geolocation of the individual traces.* We derived positioning information from dual-frequency GPS units mounted on the suspended radar frame. These were backed up in case of failure by single-frequency GPS data from GoPro cameras also mounted on the frame. The radar traces were time-stamped, so the first processing step was to interpolate GPS positions recorded every second to provide UTM coordinates for each trace on the radargram. We processed the dual frequency GPS data using the online Precise Point Positioning service of the Canadian Geodetic Survey.

b. *Noise suppression through frequency filtering.* Noise arose from within the radar system and from the environment. The main internal noise source was a low frequency radar "wow" component induced in the input stage by the arrival at the radar receiver of the high amplitude spike direct from the transmit antenna. The main environmental noise was relatively low amplitude random noise from HF radio transmissions in the region. To suppress noise, we applied a band-pass frequency filter with a 100 % pass band between 4 and 10 MHz and frequency cut-offs at 2 and 20 MHz.

c. *Amplitude adjustment.* We applied a standard, time-varying amplitude adjustment (divergence compensation) to compensate for the spherical divergence of the propagating radar wave.

d. *Trace interpolation.* We acquired the radar data with a fixed time interval between traces but as the helicopter flight speed was variable, the horizontal spacing between traces also varied. We interpolated the radar data to a fixed horizontal trace spacing as pre-conditioning for the migration step.

e. *Migration.* We employed a finite-difference migration to re-focus hyperbolic returns from point-like scatterers to their origin point. Debris-covered mountain glaciers are a complex of 3D radar targets from along the flightline and from off-axis surface features, subsurface boulders and other irregularities (see Section 2.2.2), but only 2D-migration (along flightline) was possible in this case because our profiles were spaced widely apart to improve survey coverage. Migration of cross profiles was generally more successful than for long profiles at refining the detail of glacier beds, because they tend to traverse the relatively irregular, high relief flanks of glacier bedforms that are otherwise smoothly streamlined in long profile. In the absence of 3D processing, migration was less successful at cleaning the profiles of hyperbolic events arising off to the sides of the flight line from, for example, boulders and surface cliffs. This prompted the need for further declutter processing (Section 2.2.2).

f. *Frequency-wavenumber (fk) filtering to suppress sidewall echoes.* Prominent, diagonally dipping linear reflections are apparent in many cross-profile radargrams, the result of the platform's steady approach towards, and retreat from, steep valley sides and lateral moraine banks to front and rear. Such features have a fixed wave velocity across the record which, when transformed to frequency/wavenumber space, fall in a separate region from horizontal or hyperbolic events. We were able to use this distinction to filter out many such sidewall echoes before transforming the data back to the time/distance domain.

g.   *Spatial averaging filter.* As the aim of this survey was to establish the thickness of glaciers in the region, the
targets of greatest interest were the spatially continuous beds of the glaciers. To enhance these continuous
reflections and suppress random scattering in profiles where the location of the bed was not already clear, we
applied a spatial averaging filter.

h.   *Band-pass frequency filter.* All processing steps tend to introduce noise, so to clarify the profiles we applied a
further band-pass filter after all other filtering, with pass band 2-8 MHz and cut offs of 1 and 16 MHz.

i.   *Geometry correction.* We corrected for variations in altitude of the helicopter and radar platform using the
GPS-derived positions and a radar airwave speed of 0.3 m ns$^{-1}$.

Processing steps *a-h* yielded radargrams like that in Figure 3a, which we used to identify prominent linear
reflecting horizons with a form and depth consistent with being the glacier bed. In some cases, we could identify
and digitise a single, unambiguous bed horizon, but in other cases, multiple candidate bed horizons were present
(e.g., Figure 3b) that required further analysis to distinguish the bed from clutter (Figure 3c and Section 2.2.2).
Geometry correction (processing step *i*) then placed the radargram and picked bed in their true geographical
context (e.g., Figure 3d).

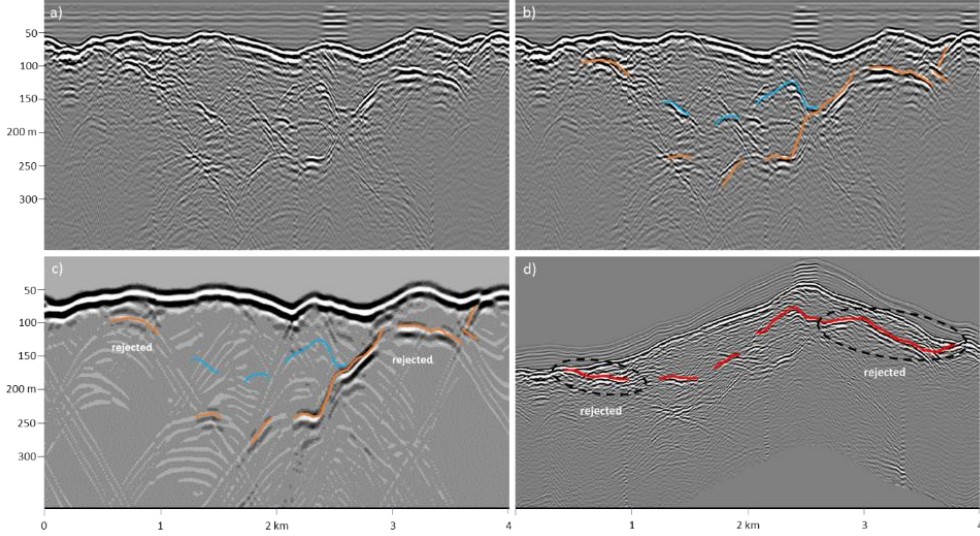


**Figure 3 |** Testing and correction of a manually picked glacier bed using output from the clutter model. a) A
survey radargram from Nuptse Glacier, Nepal, with a characteristically complicated set of bright, semi-continuous
linear features appearing at various ranges beyond the glacier surface; b) plausible bed picks (orange and blue
lines) with ambiguity where they overlap; c) the clutter model for this survey showing that the orange lines
correspond to terrain clutter but the blue lines do not, suggesting that the blue lines represent the bed; d) the
terrain-corrected radargram with the picked bed (red lines), highlighting the sections that were initially picked but
then rejected following the clutter test (panel c) (vertical scale differs from other panels) (see also Figure 6).



### 2.2.2 Clutter modelling

The unfocussed, near-isotropic nature of our dipole radar antenna pattern combined with the extreme topographic
relief in our survey area and the presence of potentially highly radar-reflective surfaces (such as smooth rock
walls, ponds and wet ice cliffs) mean that our survey data contain considerable clutter from the landscape surface,
in addition to signals from glacier beds. In particular, the lateral valley walls and moraines of glacier troughs can
generate continuous and slowly varying linear features in survey radargrams that are similar to bed signals in form
and apparent range (Pritchard, King et al. 2020). Compounding this, signals reflecting from the bed suffer greater
attenuation than surface clutter due to radar absorption and scatter by the glacier ice and so are typically weaker.
These factors can lead to considerable ambiguity when attempting to pick the bed (Pritchard, King et al. 2020)
(e.g., Figure 3b).

To help distinguish bed signals from clutter, we developed a clutter model that produces synthetic radargrams
based solely on surface scattering, using the survey flightline GPS tracks projected within a digital elevation model
(DEM). The clutter model calculates slope and aspect from the DEM and reproduces the range and incidence
angle of a spreading radar pulse to all landscape features within radar line of sight (the radar viewshed) from each
survey point, outputting the estimated amplitude of the returning signal through time for these features (Figures
4, 5). It allows for testing of a variety of surface-scattering models (Lambertian/diffuse reflection, specular
reflection, and Minnaert reflection (e.g., Minnaert 1941, Phong 1998), Fig. A1) according to the assumed
wavelength-scale roughness of the surfaces encountered, and for testing of variations in the imperfectly-known
antenna transmission pattern (from a pulse that is isotropic, to one preferentially directed towards side-lobes
perpendicular to the antenna orientation/flightline).

We used this model to approximate the continuously varying surface clutter along each survey flightline, using
model parameters that led to the closest visual match to the survey radargrams. We then visually compared the
synthetic and real radargrams (e.g., Figures 3a, c) to help distinguish between bed signals and clutter. We manually
picked the glacier beds to the best of our ability and used our synthetic clutter radargrams to test these picks, i.e.,
to identify 'false beds' that we had misidentified, which we then rejected (e.g., Figure 3d). After this clutter-
removal step, we corrected the radargram geometry to account for flying height and speed and projected the
geographical locations of the picked bed profiles (e.g., Figure 6).



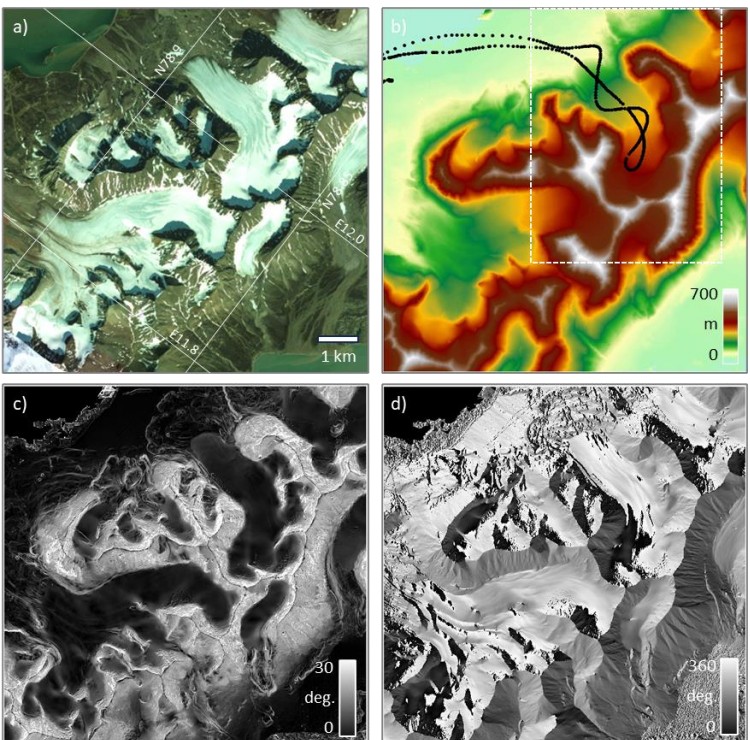

**Figure 4 |** Example of the terrain properties of a glacial catchment (a) that are used to model clutter, including: b)
elevation; c) slope; and d) aspect. The black dotted line in (b) shows a survey flightpath through the landscape
(Pritchard, King et al. 2020), the white box shows the area covered in Figure 5.

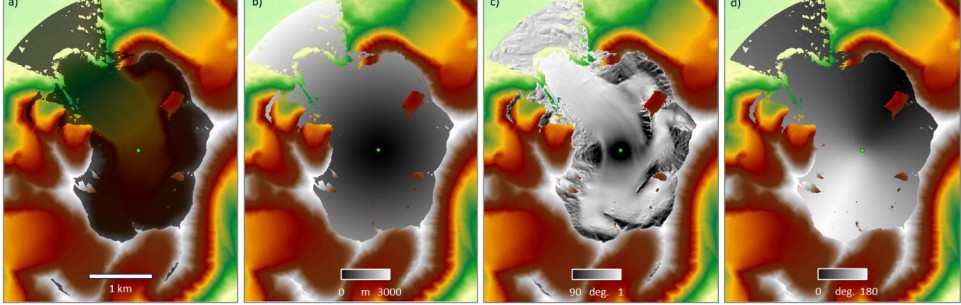

**Figure 5 |** Terrain analysis for a point (green dot in centre) along the survey flightpath in Figure 4b, including: a)
the line of sight viewshed (dark grey mask), and within this viewshed; b) the distance to terrain (greyscale); c)
incidence angle of the wavefront meeting the surface (greyscale); and d) directivity angle (flightpath orientation
relative to the surface) (greyscale). The background colour scale shows topography as in Figure 4b.

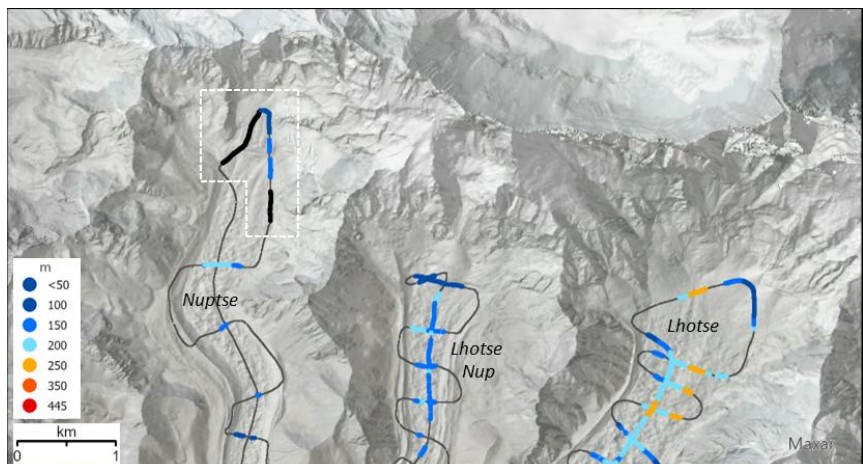

**Figure 6 |** Flightlines (grey) over Nuptse, Lhotse Nup and Lhotse glaciers, with picked ice thicknesses (colour scale) and mis-picked sections of bed (black) that were rejected following the clutter modelling shown in Figure 3 (sections in white box above). The legend shows the upper value of each thickness class in metres. Background: shaded-relief topography (Jarvis, Reuter et al. 2008, Shean 2017).

## 3 Results

### 3.1 Coverage

After data processing, we are able to report ice thickness along a total of 119 line-kilometres in this survey, sampling Khumbu Himal glaciers that cover a total area of 240 km$^2$ (RGI Consortium 2017) (Figure 7). This is a substantial improvement in our knowledge of glacier thicknesses in this region. There were previously only ~8.4 line-km of thickness data from ground surveys in Khumbu Himal (see Section 3.2) and, more broadly, our ten days of airborne surveying have approximately doubled the combined length of all surveyed profiles from all 95,000 glaciers of HMA collected over the last 60 years, many of which were concentrated on the small (median 2 km$^2$), thin (mean 52 m) and clean glaciers of the northern HMA, and predate accurate GPS survey-control so have poorly constrained locations (GlaThiDa Consortium 2020).

We measured ice thicknesses of up to 445 m at surface altitudes spanning 4670 m to 6311 m. Our measurements come from a variety of settings, from relatively clean, thin and cold glacier accumulation areas, heavily crevassed icefalls, thick and debris-covered central glacier trunks, and heavily debris-mantled, pond-covered lower glacier tongues (Figure 7). Over the thickest ice we were able to pick the bed only where the glacier surface was relatively clean, in contrast to the debris mantled glacier either side of a clean-ice band (Figure 8), demonstrating the impact of debris cover in limiting radar penetration in such settings.

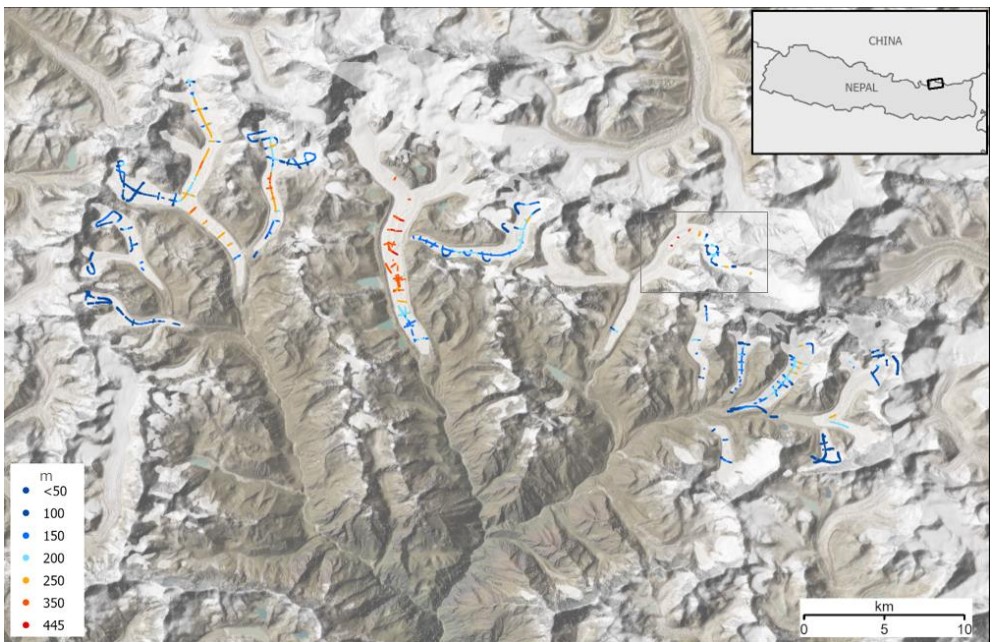

Figure 7 | Ice thickness survey results, with the thickest ice (red) in the central Khumbu Glacier (grey box and Figure 8) followed by central Ngozumpa Glacier (locations in Figure 2). The legend shows the upper value of each thickness class in metres. Background: shaded-relief topography (Jarvis, Reuter et al. 2008, Shean 2017) and glacier extents (RGI Consortium 2017) (white polygons) overlaid on satellite imagery (Earthstar Geographics via ArcGIS Online).

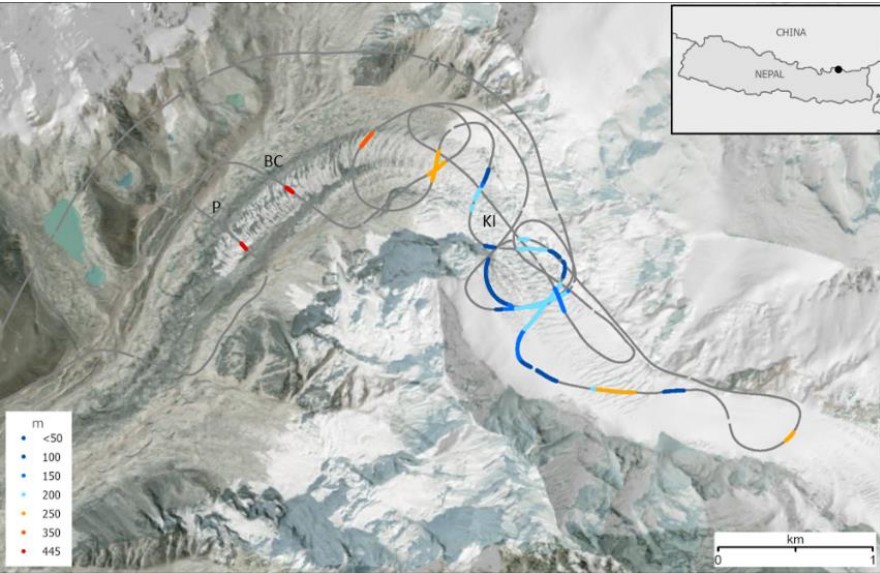

Figure 8 | Successful bed picks on Khumbu Glacier (grey box in Figure 7) showing the thickest ice through which we detected the bed (up to 445 m) in a band of largely clean, debris free ice in the central Khumbu Glacier trunk





below the Khumbu Icefall (KI). Note that this coincides closely in location and thickness with radar cross-profiles (in the vicinity of labels P and BC) from a previous ground-based survey (Gades, Conway et al. 2000). Flightlines

are shown in grey. The legend shows the upper value of each thickness class in metres. Background: shaded-relief topography (Jarvis, Reuter et al. 2008, Shean 2017) overlaid on satellite imagery (Earthstar Geographics via ArcGIS Online).

| Glacier | Survey length (picked bed) (km) | Thickest ice (m) | Mean thickness (SD) (n) | Survey minus Millan mean (SD) (n) ($\epsilon$) | Survey minus Farinotti mean (SD) (n) | Survey minus Rowan mean (SD) (n) |
|---|---|---|---|---|---|---|
| Chhule | 7.08 | 122 | 64 (32) (5108) | -27 (41) (4,456) (43) | -10 (24) (4,692) | NA |
| Melung/Pangbug | 6.56 | 187 | 95 (39) (5808) | -28 (40) (5,545) (47) | 15 (33) (5,699) | NA |
| Bhote Khosi/Lunag/Nangpa | 21.08 | 292 | 153 (73) (20940) | 4 (49) (20,799) (51) | -20 (59) (20,876) | NA |
| Lumsamba | 15.84 | 306 | 146 (69) (15815) | -7 (51) (15,187) (51) | 13 (42) (15,323) | NA |
| Ngozumpa | 31.58 | 384 | 179 (84) (21618) | -6 (68) (21,091) (57) | 0 (68) (21,395) | NA |
| Khumbu | 5.26 | 445 | 155 (83) (3051) | -12 (53) (2,795) (55) | 50 (63) (2,890) | 18 (85) (3040) |
| Nuptse | 1.59 | 174 | 129 (36) (1526) | -17 (17) (1,501) (51) | 40 (17) (1,511) | NA |
| Lhotse Nup | 3.74 | 188 | 121 (40) (2772) | -2 (28) (2,618) (48) | 25 (29) (2,667) | NA |
| Lhotse | 13.08 | 237 | 137 (49) (9201) | -55 (56) (8,406) (59) | 33 (31) (8,554) | NA |
| Imja/Lhotse Shar/Ambulapcha | 12.44 | 234 | 74 (56) (8745) | -12 (44) (6,608) (43) | 8 (25) (6,797) | NA |
| Ama Dablam | 0.66 | 155 | 105 (25) (1131) | 4 (15) (1039) (44) | 6 (19) (1,092) | NA |
| **All** | **119 km** | **445** | **139 (75) (98,984)** | **-11 (56) (93,304) (52)** | **5 (54) (94,760)** | **NA** |

**Table 1** | Ice thickness survey statistics by glacier, and a comparison to previous model results termed 'Millan'
(Millan, Mouginot et al. 2022), 'Farinotti' (Farinotti, Huss et al. 2019) and 'Rowan' (Rowan and Egholm 2021), where available (Section 3.3). The difference statistics ('Survey minus Millan', etc) are for all survey points that overlap with these model products, and report the mean difference (m), standard deviation (SD) (m), number of survey points (n) and, in the case of the Millan product, the quoted mean uncertainty for the modelled thickness ($\epsilon$) (m). (Note that as these statistics refer to the values at the set of points that we successfully surveyed for each
glacier (Figure 7): "mean thickness", for example, does not equate to the mean thickness of the entire glacier).

### 3.2 Resolution, precision, accuracy and validation

With the combination of our 3 kHz radar pulse repetition frequency, the average flying speed while surveying of ~10 m s$^{-1}$ (36 km h$^{-1}$) and the trace stacking and horizontal interpolation of our data processing, the horizontal sampling in our survey averages 1.15 m (max 3.0 m, SD 0.27 m) along flightlines. However, the horizontal
resolution of the bed target is limited by the Fresnel zone (effective radar footprint) of our transmitted pulses. At a frequency of 7 MHz and with the typical range of the glacier bed from the radar (~100-600 m, mean of ~160 m), the Fresnel zone has a radius of approximately 30-80 m (mean ~40 m).

In the vertical, the range resolution of the picked surface and bed is nominally a quarter wavelength (~7 m), but
the 'optimal vertical resolution' (the resolvable range to a single discrete, prominent reflector (King 2020)) is ~1 m, and the 'practical vertical precision' for such horizons is ~2 m at this frequency (Pritchard, King et al. 2020).

This implies that the practical vertical precision in thickness is the combination in quadrature of these two precisions, i.e., ~2.8 m.

The absolute accuracy of the thickness is subject to the accuracy of our assumed radar velocity in ice (0.168m ns$^{-1}$) with which we convert two-way radar travel times to ice thicknesses, and this is somewhat dependent on the unknown and potentially variable depth-averaged glacier water content. A range of velocities from 0.165 to 0.172 m ns$^{-1}$ has, for example, been employed for temperate and cold ice above and below the equilibrium line of an alpine glacier (Macheret, Moskalevsky et al. 1993). This range of velocities implies a difference in ice thickness

of approximately ±2 % for our survey (equivalent to a change in mean thickness from 139 m when using a velocity of 0.168 m ns$^{-1}$ to between 137 m and 142 m for the reasonable range of velocities). Given the dependence on water content, this could be manifest as a thickness bias that varies broadly with altitude, with our results potentially too thick by up to 2 % at lower (warmer) altitudes, too thin by up to 2 % at higher (colder) altitudes. Potentially more significant bias (e.g., tens of metres) could result from mistaking a non-bed reflection horizon

for the bed, which we sought to avoid with our clutter modelling (Section 2.2.2).

To assess the consistency of our picked thickness measurements, we quantified the difference in thickness at 79 flightline crossovers (Figure 9). The mean absolute crossover difference was 9 m and the mean relative difference was 7 % of thickness (median 6 m and 5 %) (Table 2). We also compared our airborne survey results to previous

ground-based radar surveys on Khumbu (Gades, Conway et al. 2000) and Ngozumpa glaciers (Pritchard, King et al. 2020). On Khumbu Glacier, seven radar cross profiles were surveyed in 1999 over the glacier tongue below the Khumbu Icefall, totalling 3.3 km in length (Gades, Conway et al. 2000). Of these, two lines crossed within ~200 m horizontally of two of our successfully surveyed cross profiles (around P and BC in Figure 8). While the earlier ground-based survey achieved profile lengths of ~500 m each, spanning most of the glacier width, we were

only able to pick the bed over around 70 m of our profiles at each location. Although these surveys differ in date, method and exact location, the thicknesses reported by both surveys are similar: we measured maximum thicknesses of 445 m close to line P and 440 m close to line BC, compared to maxima of ~370 ± 20 m (line P) and 440 ± 20 m (line BC) in the previous study (Pritchard, King et al. 2020). A thickness of ≤450 ± 70 m close to these lines was also observed by a terrestrial gravity survey in 1976 (Moribayashi 1978).

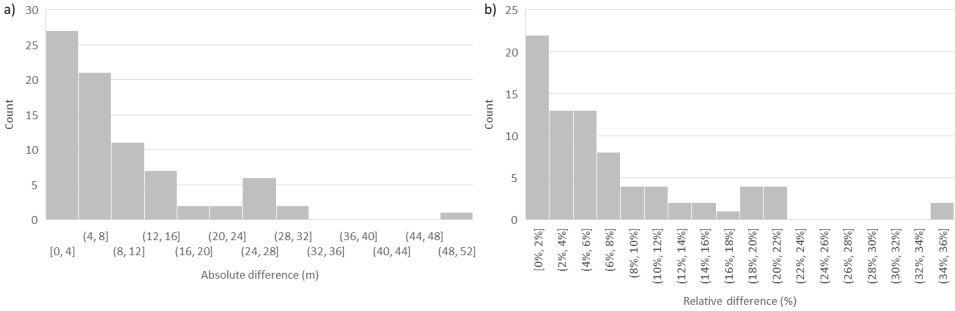


**Figure 9 |** Frequency distribution of a) absolute and b) relative crossover thickness differences.



| Crossover summary | Mean | SD | Median | Max |
|---|---|---|---|---|
| Relative difference (%) | 7 | 7 | 5 | 30 |
| Absolute difference (m) | 9 | 9 | 6 | 51 |

**Table 2** | Statistics of surveyed thickness differences measured at 79 flightline crossovers.


On Ngozumpa Glacier, we also compared our airborne survey results to an earlier ground-based radar survey that collected 5.1 line-km of data in 2016 (Pritchard, King et al. 2020). We used the ground survey in some cases to help avoid erroneous bed picks in the airborne survey radargrams, where clutter 'horizons' made bed detection ambiguous, but the thickness distributions of these datasets are otherwise independent. For four zones where both

surveys have extensive, though non-identical, coverage (Table 3, Figure 10), this comparison shows close similarity between both the means (-13 m to +8 m difference, with a weighted mean difference of ~0.2 m) and standard deviations (2-7 m difference) of thickness in these zones. The variation in sign of the thickness differences between these datasets suggests that there is no systematic bias between the surveys.

In summary, this assessment suggests that our survey data suffer from little systematic bias (typically <2 %) and have a precision that we estimate as around ±7 % of thickness (±10 m for the mean thickness of 136 m). Local inaccuracies can reach ~30 % of ice thickness.

| Data source | Mean thickness (m) | Standard Deviation | n | Thickness difference (m)(%) |
|---|---|---|---|---|
| Zone 1 - ground | 345 | 28 | 612 | 0 |
| Zone 1 - air | 345 | 21 | 578 | |
| Zone 2 - ground | 258 | 13 | 471 | +8 (3 %) |
| Zone 2 - air | 266 | 15 | 1727 | |
| Zone 3 - ground | 174 | 15 | 133 | 0 |
| Zone 3 -air | 174 | 13 | 475 | |
| Zone 4 - ground | 140 | 12 | 262 | -13 (10 %) |
| Zone - air | 127 | 10 | 996 | |

**Table 3** | Comparison of thickness statistics between samples of ground-based (Pritchard, King et al. 2020) and

airborne survey results from zones 1-4 on Ngozumpa Glacier (Figure 10). (Note that we combine the results from this ground survey with those from our airborne survey in our Khumbu Himal thickness dataset (DOI)).

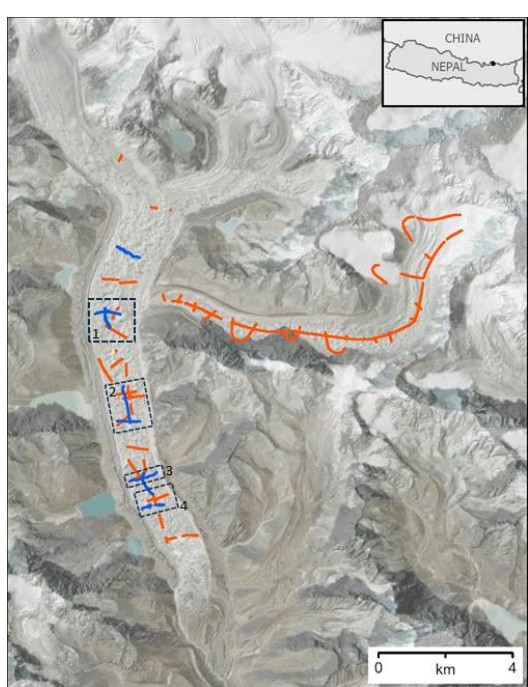

**Figure 10 |** Boxes 1-4 show the samples of ground-based survey results (blue) and the samples of airborne survey results (orange) on Ngozumpa Glacier (Table 3). Background: shaded-relief topography (Jarvis, Reuter et al. 2008,
Shean 2017) overlaid on satellite imagery (Earthstar Geographics via ArcGIS Online).

**3.3 Comparison to prior modelled thickness estimates**

Our main goal in developing our method and conducting this survey was to provide an improved observational dataset for testing ice thickness models. To demonstrate the utility of our new dataset to the broader glacier modelling community, we here make an initial, direct assessment of the performance of three existing modelled
thickness products that coincide with our survey dataset:

i)    'Rowan' (Rowan and Egholm 2021, Rowan, Egholm et al. 2021), a thickness grid for Khumbu Glacier representing the year 2011 and calculated using a flow law for plastic ice, observed surface slope and estimated basal shear stress, modified by a semi-empirical 'shape factor' accounting for the effect of the valley sides on ice flow and a down glacier 'thinning factor' to account for non-steady-state mass balance.
This relationship was tuned using the earlier radar and seismic thickness observations (Moribayashi 1978, Gades, Conway et al. 2000) (Section 3.2) and the model was able to reproduce past and present glacier extents, present surface elevations and present surface flow rates, but thickness accuracy and precision are not reported (Rowan and Egholm 2021).

ii)    'Millan' (Millan, Mouginot et al. 2022), a set of thickness grids for most of the world's glaciers for years
2017/2018 that employed the shallow-ice-approximation (SIA) to calculate thickness distribution using the observed glacier surface slope and flow rate, with region-averaged ice rheology and basal sliding parameters estimated through a regional calibration against ice thickness measurements available at that time. Accuracy and precision assessed against independent measurements from the Alps were estimated as





–16 ± 51 m, or a precision of 25–35 % for ice thickness greater than 100 m, >50 % for ice thicknesses

below 100 m. Limitations ascribed to this model include temporal mismatches between mappings of velocity (2017-2018), surface slope (2000-2015), glacier extent (2003), and ice thickness (variable dates and limited extent), and the model was found to perform relatively poorly where the surface slope of the ice is nearly flat (Millan, Mouginot et al. 2022).

iii) 'Farinotti' (Farinotti, Huss et al. 2019), a set of thickness grids for most of the world's glaciers for
approximate years 2000-2010 based on an ensemble of up to five thickness models: a) a mass conservation approach driven by a regional estimate of the surface mass balance gradient and observed surface slope and area, with regional flow parameters calibrated using thickness observations where available; b) an empirical relationship between the altitude range of the glacier and average basal shear stress within altitude zones, modified by an empirical shape factor, with further parameter calibration against available
thickness observations; c) a mass conservation approach driven by gridded climate data (localised rather than a regional estimate) and observed surface slope using SIA along multiple flowlines, also with flow parameters calibrated using available thickness observations; d) a mass-conservation and SIA approach driven by surface mass balance constrained by satellite gravimetry, altimetry and field observations, with flow parameters again calibrated using available thickness observations; e) an alternative version of (b)
above, but with only the shape factor calibrated using thickness observations. Previous tests against independent ice thickness measurements showed that the consensus thickness grids had negligible bias and a precision of ±26 % on the regional and global scale, but uncertainties were not provided for individual glaciers (Farinotti, Huss et al. 2019).

As is apparent from the model descriptions above, the accuracy of these gridded ice thickness products is dependent on the availability, accuracy and representativeness of surveyed thickness measurements which, as discussed, were particularly poor in this region. Assessed against the full extent of our new survey dataset, we found on average a small model thick bias (11 m) for Millan and a small thin bias (5 m) for Farinotti (Table 1). Biases are larger for individual glaciers, however, with a glacier-average thick bias of 55 m for Millan on Lhotse
Glacier, and thin biases of 50 m and 18 m on Khumbu Glacier for Farinotti and Rowan respectively (Table 1). Furthermore, the spatial distribution of thickness biases shows coherent patterns on the sub-glacier scale (Figure 11), some of which are consistent between the models. We find, for example, a thin bias in central Ngozumpa, central Khumbu, lower Imja and lower Lumsamba glaciers, and a thick bias on lower Ngozumpa, upper Lunag and upper Lumsamba glaciers (locations in Figure 2).

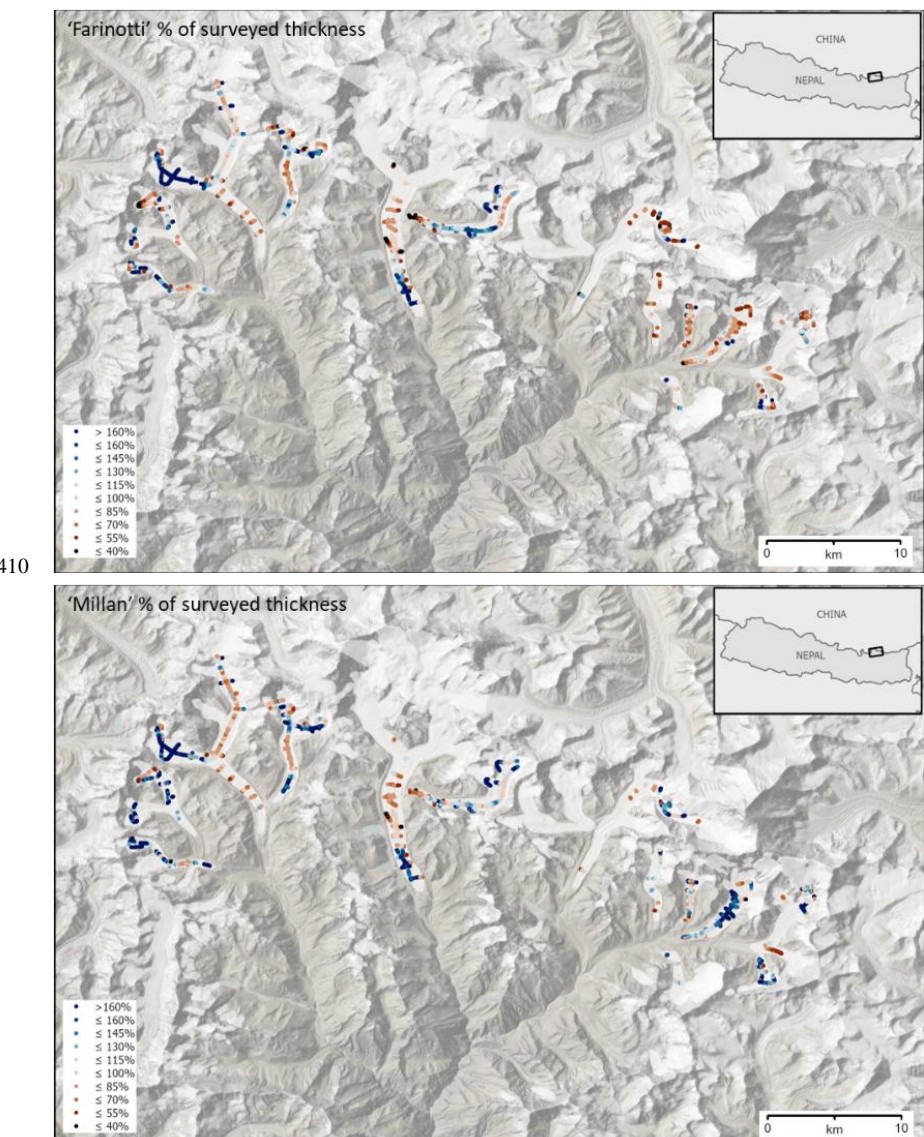


**Figure 11 |** The spatial distribution of thickness biases in the Farinotti and Millan modelled products. The values shown are modelled thicknesses as a percentage of surveyed thicknesses: blue where the model is too thick, brown where it is too thin. See also Fig. A2 for the spatial distribution of absolute biases in metres. Background: shaded-relief topography (Jarvis, Reuter et al. 2008, Shean 2017) and glacier extents (RGI Consortium 2017) (white polygons) overlaid on satellite imagery (Earthstar Geographics via ArcGIS Online).


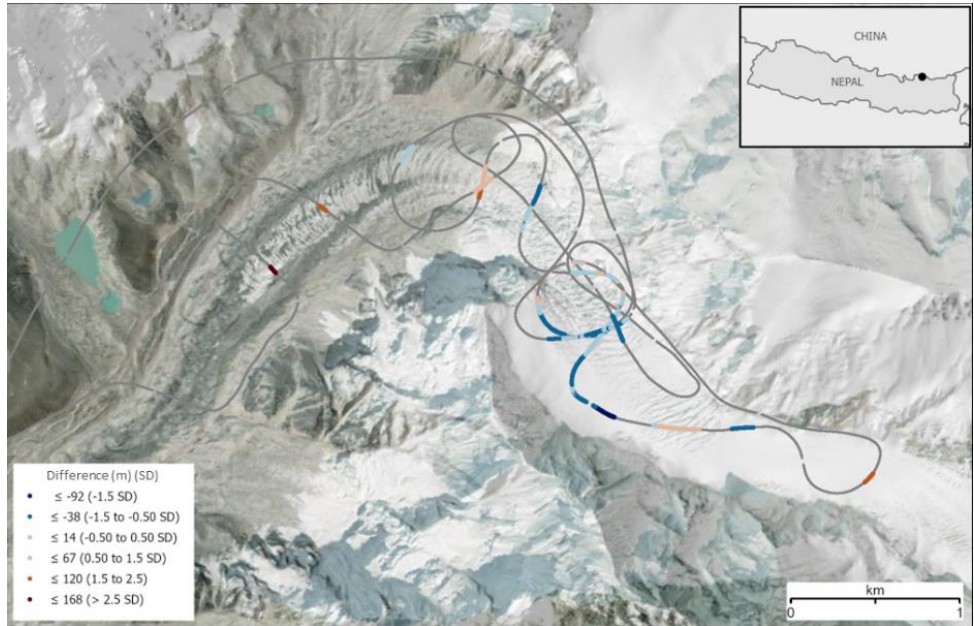

**Figure 12 |** Surveyed thickness minus modelled Millan thickness (Millan, Mouginot et al. 2022), in metres and in multiples of standard deviations (SD) of difference from the mean difference of -12 m for Khumbu Glacier.

Negative (blue) values indicate that the model grid is too thick, positive (brown) values that it is too thin. The largest thin biases in this model tend to correspond with the thickest ice and the largest model thick biases tend to correspond with the thinnest ice (cf. Figure 8). Background: shaded-relief topography (Jarvis, Reuter et al. 2008, Shean 2017) overlaid on satellite imagery (Earthstar Geographics via ArcGIS Online).

The thickness biases for both Farinotti and Millan are somewhat positively correlated ($R^2$ of 0.301 and 0.373 respectively) with absolute thickness, with both models overestimating the thickness of thin ice and underestimating the thickness of thick ice (Figures 11, 12, 13a). There is little correlation between bias and altitude when assessed over the full altitude range of the measurements, but a stronger correlation ($R^2$ of 0.376 and 0.407 respectively) for the lowest sections of the glaciers (<4900 m). Specifically, both Millan and Farinotti

overestimate the thickness of the lowest-altitude ice (<4800m) while underestimating it in the altitude range 4800-5000 m (e.g., Figure 13b) (though we note that most of our <4800 m survey data come from Ngozumpa Glacier alone (Figure 14)). We find no correlation between model thickness bias and surface slope but, for Farinotti, we find that the gridded product tends to underestimate ice thickness (by a mean of 22 m across this survey) where the glaciers flow relatively fast (>10 m a$^{-1}$), and where the flow rate has decreased most strongly

since the 1980s (by a mean of 39 m thickness for deceleration >1 m a$^{-1}$ dec$^{-1}$) (Gardner, Fahnestock et al. 2022) (Figures 14, 15). Together, these findings suggest that the models struggle to reproduce ice distribution within such Himalayan glaciers, which have complex geometries, high relief and a wide range of ice thicknesses, are not in steady state, and contain areas of near-stagnant ice in their lower tongues (Figure 14). Further analysis of individual model behaviour is needed to diagnose the causes of model bias that we reveal.


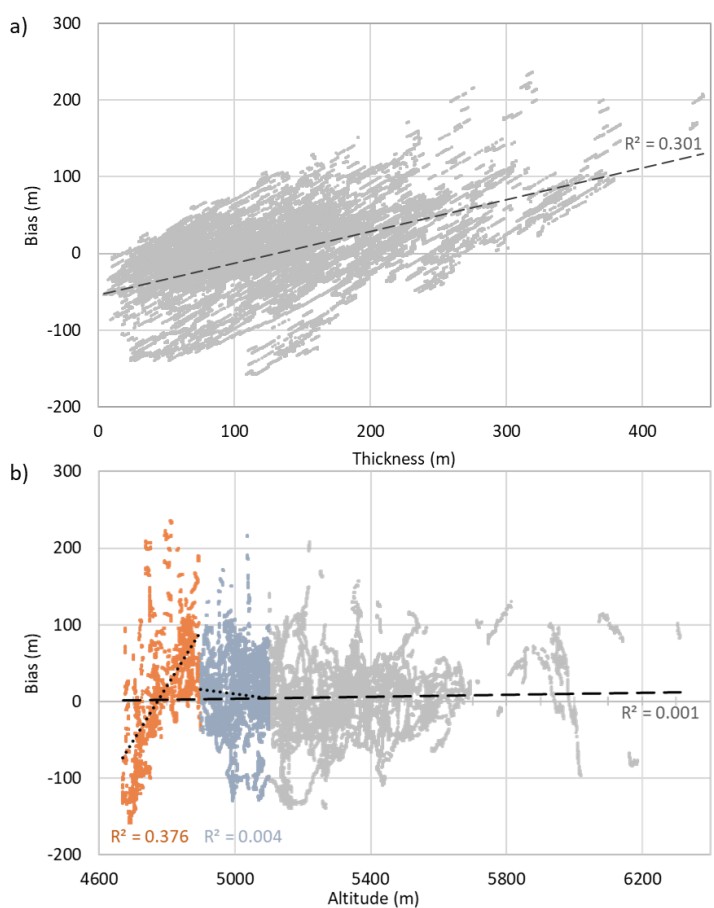

**Figure 13 |** Biases for the Farinotti thickness product plotted against a) absolute thickness, and b) altitude (Shean 2017), with coloured sections highlighting apparent variability in this relationship for different sections of the glaciers (orange = 4700-4900 m, blue = 4900-5100 m). Bias is calculated as surveyed minus modelled thickness (negative indicates that the model is too thick).

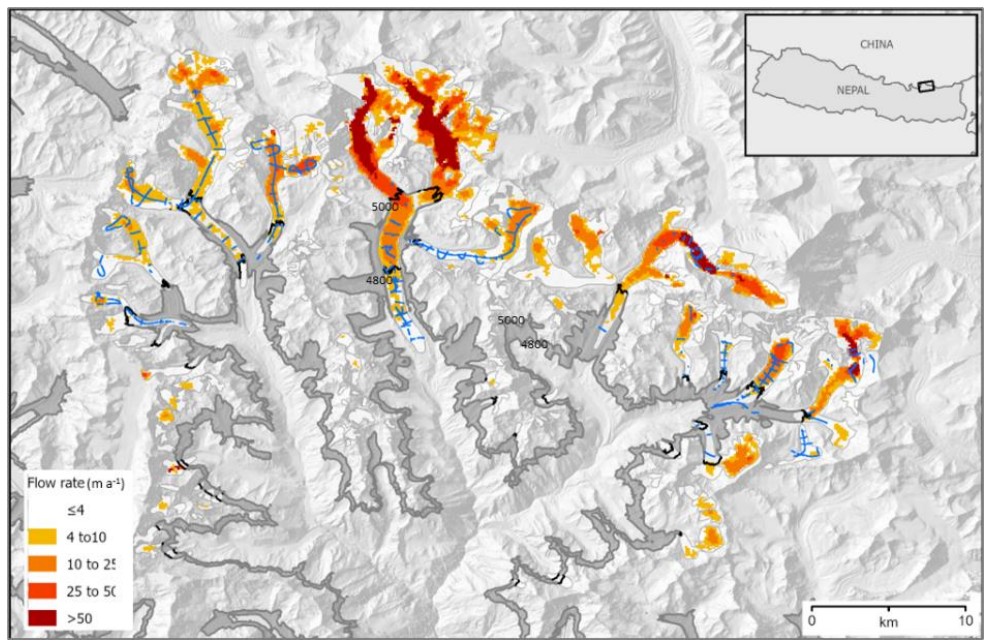

**Figure 14 |** Flow rate (after Gardner, Fahnestock et al. 2022) (colour scale) and extent (white polygons (RGI Consortium 2017)) of glaciers in this study, with the 4800-5000 m altitude band highlighted (grey polygon and black lines where this crosses glaciers) (after Shean 2017). Location of survey data is shown in blue. Background: shaded-relief topography (Jarvis, Reuter et al. 2008, Shean 2017) and glacier extents (RGI Consortium 2017) (white polygons).


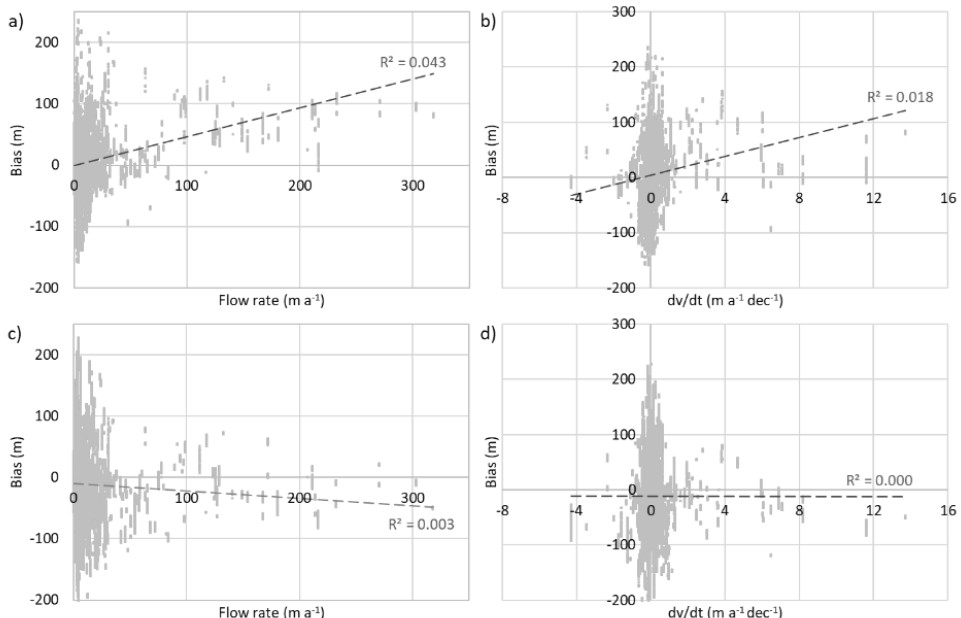

**Figure 15 |** Farinotti thickness bias against a) flow rate, and b) rate of change in flow (dv/dt in metres per year per decade) (Gardner, Fahnestock et al. 2022); Millan thickness bias against c) flow rate, and d) rate of change in flow.

**Data availability**

The original and processed radar data and the picked ice thicknesses, along with their metadata, are available from the NERC EDS UK Polar Data Centre: https://doi.org/10.5285/e39647f5-fb72-4d16-acbd-9784ed2167b8 (Pritchard, King et al. 2025).

**Code availability**

The clutter model code is available here https://zenodo.org/records/15488954 (wiki available at https://github.com/bearecinos/radar-declutter/wiki).

**Conclusions**

The portability and simplicity of our lightweight but long, modular radar platform proved well suited to helicopter surveys of remote mountain glaciers. We were able to detect the beds and map the thickness of numerous previously unsurveyed stretches of glacier in the Khumbu Himal of Nepal, including the area's largest glacier, Ngozumpa, and possibly it's thickest ice, in Khumbu Glacier, with little bias and a precision of around ±7 %. Data processing for bed detection was challenging primarily due to clutter from the surrounding terrain, but our clutter modelling proved valuable in identifying reflections that did not come from the bed. With this one survey we have doubled the length of thickness survey lines in High Mountain Asia, and our new thickness dataset provides



important insights into the performance of glacier-thickness models. It reveals, for example, that systematic model biases of tens of metres exist for some glaciers, and >100 m (and >100 % of surveyed thickness) for some sub-sections of glaciers, notably in the lower tongues of the glaciers studied. Our new survey capability, and this new survey dataset, allow the strengths and weakness of such models to be examined. This can therefore lead to

improved and validated assessments of glacier ice reserves in mountain ranges around the world, and better projections of how they will change in future.





**Appendix A**

| Processed flightline segment | Glacier |
|---|---|
| 2019_10_28_F1_P23 | Gyubanare (Ngozumpa) |
| 2019_10_30_F1_P64 | Ngozumpa |
| 2019_10_30_F1_P74 | Ngozumpa |
| 2019_11_03_F1_P32 | Lhotse |
| 2019_11_03_F2_P75 | Lhotse Shar/Imja/ Ambulapcha |
| 2019_11_04_F1_P24 | Lhotse Nup |
| 2019_11_04_F1_P56 | Nuptse |
| 2019_11_04_F1_P64 | Ama Dablam |
| 2019_11_04_F1_P75 | Nuptse |
| 2019_11_04_F1_P85 | Lhotse Nup |
| 2019_11_04_F1_P95 | Ama Dablam |
| 2019_11_05_F1_P24 | Ngozumpa |
| 2019_11_05_F1_P37 | Ngozumpa |
| 2019_11_05_F1_P45 | Ngozumpa |
| 2019_11_05_F1_P53 | Ngozumpa |
| 2019_11_05_F1_P83 | Ngozumpa |
| 2019_11_05_F1_P93 | Ngozumpa |
| 2019_11_06_F1_P15 | Bhote Kosi |
| 2019_11_06_F1_P19 | Lunag |
| 2019_11_06_F1_P35 | Nangpa |
| 2019_11_06_F1_P39 | Lunag |
| 2019_11_06_F1_P45 | Nangpa |
| 2019_11_06_F1_P49 | Bhote Kosi |
| 2019_11_06_F1_P66 | Nangpa |
| 2019_11_06_F1_P77 | Nangpa |
| 2019_11_06_F1_P87 | Nangpa |
| 2019_11_06_F1_P95 | Nangpa |
| 2019_11_06_F1_P98 | Nangpa |
| 2019_11_06_F2_P15 | Pangbug |
| 2019_11_06_F2_P45 | Pangbug |
| 2019_11_06_F2_P54 | Lumsamba |
| 2019_11_06_F2_P62 | Melung/Pangbug |
| 2019_11_06_F2_P75 | Lumsamba |
| 2019_11_06_F2_P83 | Lumsamba |
| 2019_11_06_F2_P93 | Chhule |
| 2019_11_06_F2_P94 | Chhule |
| 2019_11_06_fligh3_P12 | Khumbu |
| 2019_11_06_fligh3_P69 | Khumbu |

**Table A1 |** Table of survey flight details. The flightline names are in the format yyyy_mm_dd_flightnumber

_ProcessingStep and correspond to the processed radargram files (see https://doi.org/10.5285/e39647f5-fb72-4d16-acbd-9784ed2167b8). The glacier locations are shown in Figure 2.



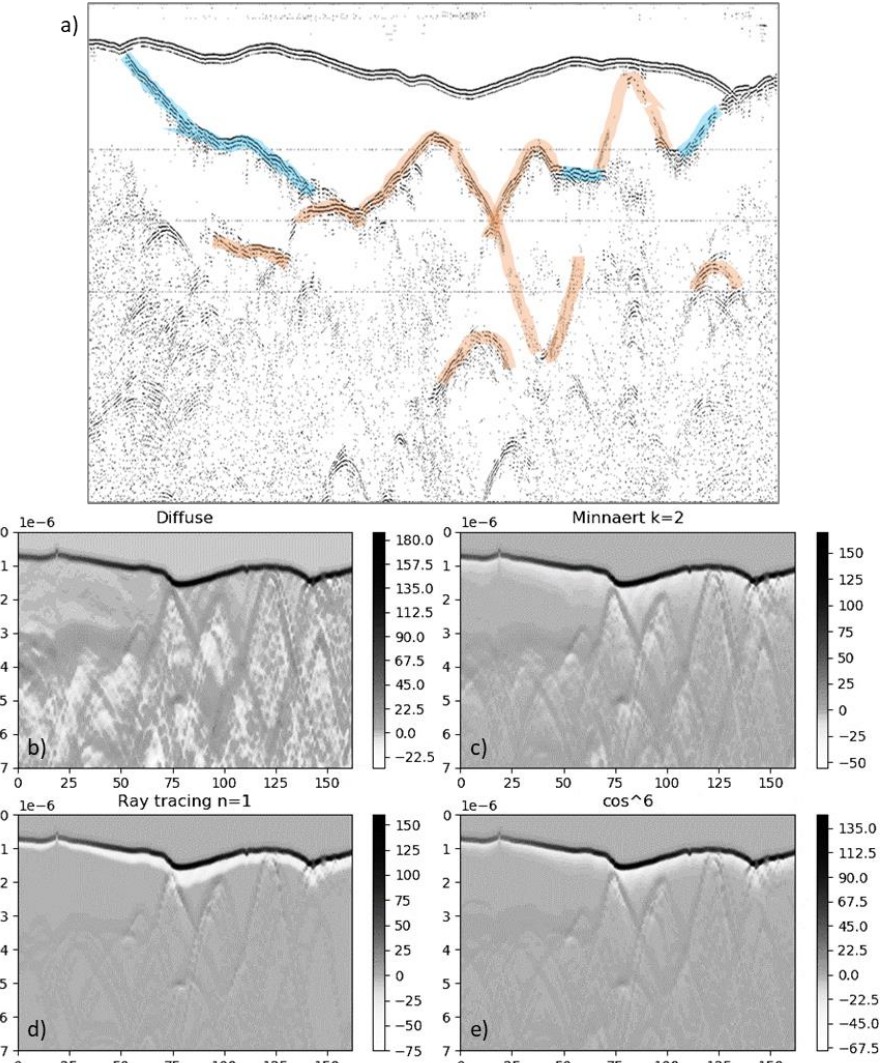

**Figure A1 |** Testing of scatter models. a) Contrast enhanced survey radargram from the flightline in Figure 4b.
(b-e) Modelled clutter using surface-scatter models (e.g., Minnaert 1941, Phong 1998) specified as diffuse,
Minnaert (k=2), specular reflection ("Ray tracing") and Minnaert ($\cos^6$) for the same flightline. Blue lines in (a)
are bed-like features absent in the modelled clutter in (b), (c), (d) and (e), and so are interpreted as bed. Orange
lines in (a) correspond to bed-like features modelled as clutter and so are not picked as bed. Greyscale values
show relative signal strength.

**Figure A2 |** The spatial distribution of thickness biases (m) in the Farinotti and Millan models. The values shown are survey thicknesses minus model thicknesses, where negative (blue) implies that the model is too thick, positive (brown) that it is too thin. Background: shaded-relief topography (Jarvis, Reuter et al. 2008, Shean 2017) and glacier extents (RGI Consortium 2017) (white polygons) overlaid on satellite imagery (Earthstar Geographics via ArcGIS Online).



**Author contribution**

HP designed and managed the project and led the field survey and manuscript preparation, EK and HP processed
and interpreted the radar data,  EK and DG designed and built the radar electronics and software and operated the
radar in the field, DB wrote the clutter tool software, DNG and BR updated and implemented the clutter tool, AO
supported the survey permitting process and DR supported the field survey design and led the field logistics
provision.

**Competing interests**

The authors declare that they have no conflict of interest.

**Acknowledgements**

This work was funded by NERC grants NE/X005267/1, NE/R000107/1 and British Antarctic Survey (BAS)
capital funding. The radar frame design and construction were led by former BAS engineer Nic Mounteney. The
Himalayan Research Expedition (P) Ltd. provided the field logistics.

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
