# Peer review of "Towards Bedmap Himalayas: a new airborne glacier thickness survey in Khumbu Himal, Nepal."

_Earth System Science Data, 2025_

## Author Response (AR1)

**Reviewer 1**

Many thanks Howard for your supportive review and helpful comments. Please see below our responses to the specific suggestions:

1) Fig. 2: Glacier names are small – perhaps put them with a white background??

Thanks, we have now increased the font size and added a grey background to help legibility.

2) Fig. 3: Clutter modelling appears to be working well. However, it is not clear that the clutter test (panel c) (vertical scale differs from other panels) – should it refer to the vertical scale differs for (panel d), rather than (panel c)? Or am I missing something?

We have now clarified this to read:

"…(panel d vertical scale differs from other panels)…"

3) Fig. 4: Use black font for (panel b) to make it visible

Agreed, we've now made this change.

4) Fig. 7: Make the lines around the "grey box" thicker to illustrate the central Khumbu Glacier

Also now done.

5) Fig. 14: Inset (bottom left) is cropped (on RHS)

Thanks, now corrected.

**Reviewer 2**

Many thanks Thomas for your thoughtful review and comments. Please find below our responses to specific suggestions.

1) Section 2.1.1: It would be useful to include the transmit power of the radar system

Good point. The actual transmitted power of full systems like this is difficult to determine and instead is typically specified as the transmitter voltage, which is known. We have therefore changed this sentence to:

"We used a wide band mono-pulse dipole radar with a transmitter that produces ±2000 V pulses at a centre frequency of 7 MHz (Pritchard, King et al. 2020) for this survey"

2) Figure 3 (d) is confusing to me. The Y axis should be shown separately for this subplot if it needs to be different from the others. It's also confusing that the areas marked as rejected have a red line through them apparently indicating a bed pick.

Thanks, yes, we now show the Y-axis for panel (d) separately and for clarity have changed the caption to read:

"…d) the terrain-corrected radargram with the initial bed pick bed (red lines), highlighting the sections that were subsequently rejected following the clutter test (panel c) (panel d vertical scale differs from other panels)… "

3) Section 3.2: I appreciate the careful analysis of resolution and uncertainty for the radar system and particular survey setting. I would suggest adding a note to the second paragraph to point out that the "vertical precision" refers to the ability to detect the location of a particular target and that ambiguities in picking which target is the basal interface could significantly exceed 2.8 m.

We have changed Section 3.2, paragraph 2 to read:

"...This implies that the practical vertical precision in thickness (assuming that the bed reflector has been correctly identified (see below)) is the combination in quadrature of these two precisions, i.e., ~2.8 m".

The 'see below' refers to the next paragraph on accuracy (as opposed to precision), which notes that "Potentially more significant bias (e.g., tens of metres) could result from mistaking a non-bed reflection horizon for the bed, which we sought to avoid with our clutter modelling..."

4) Data files: It would be helpful to include timestamps in the CSV bed picks and GPS files. This is a region that will change rapidly. Hopefully more work continues to be done here. As efforts are made to unify data collected in the region, timestamps in every file make data collation and post-processing much easier

This is a good point, and in fact the bed-pick csv files have the date stamps encoded in the first column ("heli_line") for each point, at daily resolution, e.g. the first 10 characters of entry 2019_11_06_F2_P93 are in format yyyy_mm_dd. The next column ("survey") notes the approximate period of the survey in free text format, e.g., "Helicopter October/November 2019", though the period is more precisely defined in the header metadata which include standardised date flags as, e.g., "# time_coverage_start: 2019-10-27" and "# time_coverage_end: 2019-11-06" (Please see the metadata held in the header of each csv file). Similarly for the GPS data, the date is encoded in the filename, e.g., the first 10 characters of the filename 2019_10_28_F1_coords.csv are also in format yyyy_mm_dd. In using this approach, we were governed by the metadata formatting policies of the Polar Data Centre.

5) A readme file packaged with the raw data would be very helpful to help understand how to decode it.

This is an understandable suggestion as the absence of a downloadable readme file in this folder is a little confusing, though we're again constrained by the formatting policies of the Polar Data Centre. Instead, the metadata for the raw datasets are actually held with the main combined metadata record associated with our submission, available here: https://data.bas.ac.uk/full-record.php?id=GB/NERC/BAS/PDC/02073. Fortunately, there is a link to this on the page that serves the raw datasets themselves. When scrolling down on this page, the metadata for the raw data reads:

*Raw radar data *

*We collected a raw radar data file for each survey flight. The filename format gives the date and time of file creation as yyyymmddhhmmss.hdf5, e.g., 20191103021438.hdf5.*

*File contents:*

*Each file contains two channels of radar data as recorded by the digitiser in the radar receiver*

*system. Channel A is amplified, Channel B unamplified.*

*For each channel, there are multiple Traces, numbered sequentially (e.g., Trace00001). Each trace records a series of radar receiver signed voltages as a measure of returning signal strength.*

*Channel metadata:*

*The total duration of each trace (and therefore its range, in number of samples) is given by the NoOfSamples parameter in the channel metadata. Each sample has a duration of 2 nanoseconds, hence a trace with 10152 samples is 20304 ns long (a range of around 1700 m in ice, or 3000 m in air).*

*The 'SampleRate' parameter gives the timeout time (in seconds) for the receiver to wait to detect a returning pulse before assuming that triggering has failed and moving on to the next set.*

*Each trace represents the averaged voltages of the number of independent radar pulses given by the 'Stacks' parameter.*

*'Repeats' is an internal system parameter that sets the number of samples to store before writing to disk.*

*The GPS time is recorded along with the corresponding file start and end 'system' times for the onboard computer, to allow the system timestamps to be synchronised to the GPS data in post processing.*

*Trace metadata:*

*In each file, Trace00001 metadata shows the number format (Type) and the precise system date and time (yyyy,mm,dd,hh,mm,ss.ssssss). The 'GGA' string reports the GPS time at completion of this trace as hhmmss.s (e.g., 082520.8) and the GPS position at that time (ddmm.mmmmm, N/S, dddmm.mmmmm, E/W) (e.g., 2748.34269,N,08641.95129,E means 27deg 48.34269' N, 086deg 41.95129' E).*

*A new system time and GPS time and position is then stored for every 1000th subsequent trace, hence for trace 1001, 2001 etc. This is because GPS data could not be written to disk at the same rate as the trace creation. High frequency, high precision GPS times and positions were stored internally on the GPS, however, and matched to trace times in post-processing.*

6) GitHub repository: Please consider adding a license to your GitHub repository.

Good idea, we have applied the BSD-3 Clause license to the repository (https://github.com/bearecinos/radar-declutter/blob/master/LICENSE).